# SNPtotree—Resolving the Phylogeny of SNPs on Non-Recombining DNA

**DOI:** 10.3390/genes14101837

**Published:** 2023-09-22

**Authors:** Zehra Köksal, Claus Børsting, Leonor Gusmão, Vania Pereira

**Affiliations:** 1Section of Forensic Genetics, Department of Forensic Medicine, Faculty of Health and Medical Sciences, University of Copenhagen, 2100 Copenhagen, Denmark; zehra.koksal@sund.ku.dk (Z.K.); claus.boersting@sund.ku.dk (C.B.); 2DNA Diagnostic Laboratory (LDD), State University of Rio de Janeiro (UERJ), Rio de Janeiro 20550-013, Brazil; leonorbgusmao@gmail.com

**Keywords:** phylogenetic tree, evolutionary genetics, population genetics, haploid markers, non-recombining DNA, SNPs, software

## Abstract

Genetic variants on non-recombining DNA and the hierarchical order in which they accumulate are commonly of interest. This variant hierarchy can be established and combined with information on the population and geographic origin of the individuals carrying the variants to find population structures and infer migration patterns. Further, individuals can be assigned to the characterized populations, which is relevant in forensic genetics, genetic genealogy, and epidemiologic studies. However, there is currently no straightforward method to obtain such a variant hierarchy. Here, we introduce the software SNPtotree v1.0, which uniquely determines the hierarchical order of variants on non-recombining DNA without error-prone manual sorting. The algorithm uses pairwise variant comparisons to infer their relationships and integrates the combined information into a phylogenetic tree. Variants that have contradictory pairwise relationships or ambiguous positions in the tree are removed by the software. When benchmarked using two human Y-chromosomal massively parallel sequencing datasets, SNPtotree outperforms traditional methods in the accuracy of phylogenetic trees for sequencing data with high amounts of missing information. The phylogenetic trees of variants created using SNPtotree can be used to establish and maintain publicly available phylogeny databases to further explore genetic epidemiology and genealogy, as well as population and forensic genetics.

## 1. Introduction

The evolutionary history of a group of taxa can be understood and recovered by studying a set of characters, such as DNA sequences, and their evolutionary relationships. These can be represented in a phylogenetic tree, which is a tree-shaped hypothesis of the degree of sequence relatedness. The taxa are represented at the tree tips, and closely related taxa progressively converge into the internal nodes of their local most recent common ancestors (MRCAs), which ultimately converge to the root [1,2].

Phylogenetic trees that represent the hierarchical order in which genetic variants were accumulated over time can also be generated. This is possible for variants (polymorphisms) on DNA regions that do not undergo recombination, such as haploid pathogens or the human Y chromosome and mitochondrial DNA. Unravelling the hierarchical order in which variants of non-recombining DNA were accumulated over time is highly informative and has a wide range of applications in epidemiological research [3,4], genetic genealogy [5], forensic genetics [6], and characterizing the genetic evolution of entire species [7,8].

To generate a phylogenetic tree for variants on non-recombining DNA, current methods utilize distance-based and character-based approaches.

Character-based approaches are more accurate and include heuristic and probabilistic methods [9]. Maximum Parsimony (MP) belongs to the former and generates the phylogenetic tree with the fewest number of character changes, irrespective of the specific mutation from one nucleotide to another—a concept that may be too simplistic to reflect reality [2,4]. Among probabilistic methods, maximum likelihood (ML) approaches (implemented in RAxML [10], IQ-TREE [11], PhyML [12], and MEGA [13]) generate multiple phylogenetic trees, determine their likelihoods, and select the tree with the highest likelihood. Alternatively, Bayesian inference (BI) uses the likelihoods of trees to update prior information, which is iteratively used to generate a distribution of possible phylogenetic trees and to build a consensus tree. Contrary to MP, ML and BI consider branch lengths and transition rates [1,2,4].

The abovementioned methods assume similar transition rates at different sites. This oversimplification disregards site-dependent structural and functional roles, constraints, selection processes, and time-dependent changes. Using artificial neural network-based approaches to infer phylogenetic trees, when appropriate training data are available, can be a way to take this information into account [14,15,16].

The evolution of characters (i.e., genetic variants) can be extrapolated based on the phylogenetic tree generated using MP, ML, or BI. Using ancestral state reconstruction (ASR), the internal nodes of the phylogenetic tree can be annotated with the most likely ancestral states, representing the character changes over time from the tree root to the tips [1,2]. ASR is implemented in, e.g., BEAST [17], MrBayes [18], and PastML [1]. Unfortunately, current ASR methods can process only one character per tree [2,19,20,21]. For nucleotide variants, this would require one tree for each variant position [3]. Alternatively, variants exclusively shared by sequences defining a clade in the generated phylogenetic tree can be manually assigned to the respective tree branch or node, representing the order in which variants were accumulated.

ASR methods are highly dependent on the accuracy of phylogenetic trees, which is defined as the similarity between the estimated tree and the true phylogeny. The tree’s accuracy is particularly uncertain when the datasets contain high amounts of relevant missing data, which is typical for ancient or forensic samples and datasets compiled from different sources. Depending on the total number of analyzed characters and the distribution of the missing data among the characters, missing data can result in multiple taxa placements with equal probability and, potentially, in inaccurate phylogenetic trees [22,23,24,25,26].

Traditional strategies for handling missing data include the selection of robust model-based tree construction methods (ML, BI) or the filtering of missing data by excluding certain taxa or characters. The latter also removes data that might not be missing in some sequences, which can reduce phylogeny accuracies [22,25]. Machine learning-based imputation of missing data has been explored specifically for distance-based approaches [27,28]. Data imputation for character-based approaches requires a reference dataset of a comprehensive set of variants that are comparable to those in the study data to extrapolate correlations between the variants. These are used to predict missing information in the study data [29,30,31,32]. Although genomic imputation is commonly applied in genome-wide association studies, an appropriate reference dataset is not always available [29,30,31,32]. Phylogeny-based data imputation of non-recombining haploid taxa determines the correlations between variants based on the hypothesis that neighboring sites on a user-provided phylogenetic tree are identical by descent [33]. If relevant data are missing, constructing an accurate phylogenetic tree as the fundamental source of variant relationships may not be possible [33].

Here, we introduce the software “SNPtotree v1.0”, which sorts biallelic variants on non-recombining DNA into a phylogenetic tree. SNPtotree circumvents the uncertainties introduced by missing data and is applicable even for sequencing data with substantial amounts of missing information. The software further allows identification of the variants that best explain the evolutionary relationships between the investigated taxa. SNPtotree was validated on two different human Y-chromosomal massively parallel sequencing (MPS) datasets but is not restricted to this application. SNPtotree was benchmarked using a robust and accurate ML-based approach. It facilitates easy creation and maintenance of variant tree databases, renouncing error-prone manual sorting of variants into their hierarchical order. The established phylogeny of variants can be combined with information on the geographic spread of the individuals carrying the variants to reveal migration patterns of different species [34]. This helps to better understand the evolution and relatedness of species or populations [8,35,36], which may find application in genetic or forensic genealogy [37]. Further, establishing or expanding population databases is significant in forensic genetics, e.g., due to population frequency estimations [38,39]. Additionally, the spread of haploid pathogens is informative for prevention strategies in epidemiologic studies [3,4].

## 2. Materials and Methods

### 2.1. SNPtotree

The allelic information of previously called and aligned variants is entered into SNPtotree, which organizes biallelic variants into a rooted phylogenetic tree.

The Python source code with detailed guidelines for SNPtotree v1.0 is publicly available at https://github.com/ZehraKoksal/SNPtotree (accessed on 20 August 2023), where future updates and regular bug fixes will be found. SNPtotree can be run within the terminal as described below:

python SNPtotree.py path_to_input_file.csv/ path_output_folder/ -contradictory_variants -ambiguous_variants -metadata_individuals

The user needs to provide the paths for the input (path_to_input_file.csv/) and output files (path_output_folder/), while the remaining arguments are optional (-contradictory_variants and -ambiguous_variants for generating optional output files, including the variants removed due to their contradictory pairwise relationships or ambiguous positions in the tree; -metadata_individuals for generating an optional output file specifying the sequences carrying the respective variants in the different branches).

Figure 1 presents the three SNPtotree algorithm steps processing the input file to generate the output files.

#### 2.1.1. Input

SNPtotree requires a tab-separated csv file as input. This format is similar to the fundamental character-by-taxon data matrix used in phylogenetic analysis [25,40]. The input represents taxa (here: individuals) in columns and characters (here: variants) in rows. Only polymorphic sites are accepted in the input file. The header row and the first (index) column should present the individuals’ labels and the variant names, respectively (Figure 1). The SNPtotree input matrix comprises the observed allelic states (ancestral “A” or derived “D”), while missing data is denoted with an additional character (“X”). The user-provided information on the observed allelic states is obtained by comparison to an outgroup with ancestral traits (e.g., due to the reference alignment of the sequencing data). This is comparable to rooting a phylogenetic tree using ancestral outgroups. Further information and an example input file can be found at: https://github.com/ZehraKoksal/SNPtotree, accessed on 20 August 2023.

#### 2.1.2. Output Files

SNPtotree provides the hierarchical order of variants in two alternative output file formats. The more traditional phyloxml file allows annotation of nodes, branches, and tips with variant labels. This output file can be depicted in phylogenetic tree visualization tools that support this file format, such as the Interactive Tree Of Life (iTOL) [41].

The variant hierarchy is also stored in a tab-separated csv file. It adopts the tree architecture employed in the database “Y-DNA Haplogroup Tree” by the International Society of Genetic Genealogy (ISOGG). Evolutionary older variants are positioned towards the tree root (left) and younger variants towards the tree tips (right). A variant that is located closer to the tree tips than another variant of the same lineage is described as “downstream”, since it succeeds the other “upstream” variant. The divergence of a clade to downstream variants is represented by the latter occupying cells in the csv file at the bottom right. “Equal” variants cannot be separated based on their relative relationships to other variants and are stored in the same cell. “Parallel” variants (in sister clades) are presented in different cells in the same column.

Associated with the csv output tree, an (n × 2)-matrix (n = number of tree branches) can be generated in an optional metadata file. Here, the variant(s) of each tree branch are returned in the first column, and all individuals carrying at least one of the variants are listed in the second column. This file allows the user to connect the constructed tree branches and their variants with the individual sequences, providing sufficient information to correlate sequences with variants.

Statistical support values for the position of each variant represented in the tree are given in the ‘certainty_values.csv’ file. These certainty values correspond to the fraction of variants in the tree that support the variant’s position based on their pairwise relationships, if these were informative (=upstream, downstream, parallel).

For examples of all output file formats, please visit the github repository via: https://github.com/ZehraKoksal/SNPtotree, accessed on 20 August 2023.

#### 2.1.3. Algorithm

##### Pairwise Variant Comparison and Removal of Variants with Contradictory Predictions

The initial step in establishing a phylogenetic tree of variants is to determine the hierarchical order of each pair of variants. Using the example dataset presented in Table 1, it is demonstrated whether the variants M1 and M3 are located downstream/upstream of each other (Figure 2A), parallel to each other (Figure 2B), or equal/not separable (located on the same branch) (Figure 2C). To establish this relationship, the allelic states between variants M1 and M3 are compared among all sequences in a two-way pairwise comparison (Figure 2D). Firstly, all sequences in which M1 is in the derived state are considered. For these sequences, the observations in M3 (derived, ancestral, and missing data) can be used for a preliminary assessment of the pairwise relationship between the two variants. This assessment follows the rules presented in Table 2 of observable allelic states in variant 2 when considering only sequences with a derived variant 1. In the example presented, for sequences with a derived allelic state of variant M1, the allelic state of M3 is derived, ancestral, or missing, which indicates that M1 is upstream of M3 (Figure 2D, Table 2). When repeating the comparison, considering sequences with derived allelic states in M3, M1 is found in the derived state, indicating equal upstream or downstream relationships. Finally, the consensus between both assessments is that M1 is upstream of M3. For some variant pairs, however, the information may not be as clear. For example, in cases where the available data was not able to fully ascertain the hierarchy of the variants, the relationship between two variants will be characterized as “equal”. Adding more data (e.g., re-typing sequences) could potentially help to resolve the hierarchy of variants.

Based on the conditions in Table 2, two-way pairwise comparisons were conducted for all variants, and the resulting relationships between variants were analyzed to determine if the results were consistent. When inconsistencies are observed, the involved variants have contradictory relationships, which can be explained by recurrent mutations, backmutations, or sequencing errors (elaborated in Appendix A). When contradictory findings are observed, the variants causing the most contradictions are successively removed from further analyses until no contradictory findings are observed among the remaining variants and stored in the “contradictory variants” output file for manual inspection.

All pairwise variant comparisons were summarized in a table that stores each variant and all its downstream variants (Table 3). Each row in Table 3 can be considered a preliminary branch of a phylogenetic tree.

##### Combining Equal Variants and the Removal of Variants with Ambiguous Results

Variants in parallel preliminary branches are compared for overlapping information to connect clades.

Variants with identical relationships are combined and considered “equal” (i.e., located in the same branch). Variants that have several possible (thereby “ambiguous”) positions in the tree are removed.

Firstly, SNPtotree ensures that every variant occurs no more than once in the tree. If possible, SNPtotree merges parallel branches that contain shared variants by combining their preceding variants to be equal. Alternatively, the branch with lower resolution is removed to avoid ambiguity in the tree while maximizing the tree resolution. Secondly, SNPtotree guarantees that variants found in one individual are located on one branch and not on several parallel branches. Variants located at branch tips are combined into a single tree position if they are observed in derived states in the same sequence. If necessary, single variants are removed if they are in branches parallel to the most resolved branch containing variants of the same sequence. These conditions are integrated into the four possible scenarios in Figure 3, where equal variants are combined and variants with ambiguous positions in the tree are removed.

##### Generating the Phylogenetic Tree

The updated pairwise predictions are combined to infer the most parsimonious hierarchical order of the variants, from the most upstream variant (root/base) to the downstream tree branch tips. Finally, the phylogenetic tree is generated and saved in a phyloxml file and in a tab-separated csv file.

### 2.2. Datasets and Nomenclature

SNPtotree was validated using two datasets of human Y-chromosomal variants with different amounts of missing data.

The first dataset was generated from 195 individuals of the human Y-chromosomal clade “Q”, with missing data ranging from 0 to 68% untyped bases per individual [42]. A total of 22 variants were detected in the dataset. For ML tree construction, only unique sequences may be included, which reduced the dataset to 51 individuals.

The second dataset comprised 46 individuals of a clade within the human Y chromosome, called “C”. The dataset was combined from two different sources [36,43], and the percentages of missing data were approximately 65% and 20% in the respective sources. A total of 4348 variants were detected in the combined dataset.

To determine the accuracy of the generated phylogenetic trees, only single nucleotide polymorphisms (SNPs) with reported phylogenetic relationships in the ISOGG Y-DNA Haplogroup Tree 2019–2020 (Version: 15.73) were compared. Further, to make the phylogenetic relationships comprehensible to the reader, the variant names (e.g., Q L232) were labeled with the SNP names (e.g., L232) preceded by the respective branch name (e.g., Q) as defined in the database ISOGG Y-DNA Haplogroup Tree 2019–2020 (Version: 15.73). The nomenclature of the branches follows these rules: Within a clade, the naming of a subsequent branch follows the pattern of adding numbers or letters lexicographically to the name of the ancestral node, e.g., branch C1 bifurcates to C1a and C1b, and C1a splits into C1a1, C1a2, and C1a3. Variants that were not previously reported were presented by the GRCh37 positions in the Y chromosome.

### 2.3. Maximum Likelihood Phylogenetic Trees Using RAxML

ML-based phylogenetic trees were generated using RAxML v8.2.12 [10] for both datasets by transposing all polymorphic positions to a fasta file. The ML phylogenies were built using the General Time-Reversible (GTR) Model with the γ model of rate heterogeneity “ASC_GTRGAMMA” with the Lewis ascertainment bias correction to prevent an overestimation of differences between sequences when using only variable sites (Appendix A).

The resulting RAxML_bipartitions.X file that contains the input tree with confidence values (0–100) on the nodes was used to illustrate the tree in the software FigTree v.1.4.4. The trees were manually rooted in FigTree using outgroups with ancestral characters.

## 3. Results and Discussion

### 3.1. Comparing SNP Phylogeny Trees of Testdata 1 by Using SNPtotree and Maximum Likelihood (ML) Trees

The phylogenetic hierarchy of the 22 variants identified in testdata 1 is publicly available (ISOGG Y-DNA Haplogroup Tree 2019–2020) and presented in Figure 4A. The figure shows that all 22 variants belong to clade Q and are presented as twelve tree branches.

The SNP phylogeny resulting from the SNPtotree analysis of testdata 1 is presented in Figure 4B and successfully represents the hierarchy of the main branches. Of the 22 variants, 17 variants were represented in seven branches of the phylogenetic tree, which corresponds to a resolution of 58% (7 out of 12 tree branches) compared to the database. A reduced resolution was expected since the ISOGG Y-DNA Haplogroup Tree 2019–2020 is composed of more sequences, representing the current understanding of the hierarchy of these SNPs. The 17 SNPs had certainty values ranging from 0.75 to 1.0 (median = 0.88), implying overall reliability of the SNP phylogenies that corresponded to that in the ISOGG Y-DNA Haplogroup Tree 2019–2020 (Figure 4A).

The phylogeny of the sequences from testdata 1 based on ML analysis is presented in Appendix A. Phylogeny confirms that all sequences were members of clade Q1. Bootstrap (BS) values were presented in a cladogram in Appendix A. For most nodes, BS values were rather low, with a predominance of BS values of 0. The BS values increased (ranging from 7 to 100) for the split-off clades Q1a/Q1a1, Q1b1a2, and Q1b1a3. The hierarchical order of SNPs and their five tree branches (42% resolution) is presented in Figure 4C. It becomes evident that this approach fails to present an early separation of the Q1a/Q1a1 clade from Q1b. The latter should instead be split into Q1b1a1(a1i2~), Q1b1a2, and Q1b1a3. Further, the tree does not truthfully capture that the Q1b1a1a1i2~ clade has a higher resolution compared to clades Q1a/Q1a1, Q1b1a2, and Q1b1a3.

In general, the SNP phylogeny resulting from the SNPtotree v1.0 software was closer to the expected phylogeny compared to the one resulting from the ML-constructed tree. Both trees separated the sister clades Q1a/Q1a1, Q1b1a2, and Q1b1a3 (Figure 4B,C). Interestingly, these branches had rather high BS values in Appendix A, and the sequences comprising these branches had low amounts of missing data (Q1a/Q1a1 median: 11%; Q1b1a2 median: 9%; Q1b1a3 median: 11%). The correlation between the amount of missing data, BS values, and resolution of the tree branches was further investigated with a more comprehensive dataset (see below).

### 3.2. Exploring the Correlation and Extent of Missing Data and Tree Resolution Using the Comprehensive Testdata 2

Testdata 2 comprised 4348 variants, 2117 of which were reported SNPs with a known phylogenetic hierarchy and 2231 of which were novel variants.

SNPtotree separated 4071 variants (out of the 4348 variants) into 81 different branches. The phylogenetic tree was combined with the metadata output file and presented in Appendix A. For each tree branch containing a single variant or a group of equal variants, Appendix A presents the corresponding individual(s) in which at least one of these variants was found.

The phylogeny of all Y-chromosomal clade C sequences from testdata 2 based on ML tree construction is presented in Appendix A. The presence of known SNPs assigned to certain clades supported the overall tree structure. The BS values ranged from quite confident values for some clades to very low values for other clades (Appendix A). In the clade introduced by the SNP AM00694 (clade C1b1), very low BS values predominated. Interestingly, this clade consisted of sequences with a high number of missing bases (marked in red in Appendix A). These sequences had missing data of 78% instead of 20% as for the majority of the sequences. This was the consequence of combining data from different sources. The five sequences with a high amount of missing data assigned to the C1b1-AM00694 clade were misplaced in the presented branches and should be placed in parallel branches instead.

Sequences with high percentages of missing data cause low BS values in ML trees because the best ML tree is generated based on a set of aligned sequences that, by chance, will differ from those selected for the bootstrapping. When sequences contain very different information (e.g., resulting from a large amount of missing data in the sequences), the information contained in the best ML tree and the BS trees may also differ significantly. A small number or total absence of overlapping tree substructures between the best ML tree and the BS trees results in BS values that are small or 0. The consequence is an unreliable hierarchical order of variants inferred from the sequences with lower BS values. To illustrate this, clades with moderate BS values (clade C2b1a) and low BS values (clade C1b1) in Appendix A were selected. The hierarchical order of the variants in the respective clades was inferred from the ML-based subtrees and compared to the hierarchies taken from the SNPtotree-based tree in Appendix A.

For the analysis of clade C2b1a, both SNPtotree and RAxML gave the same hierarchical order of known SNPs (Appendix A), which supports the expected phylogeny of all known branches from testdata 2 (Appendix A). In other words, for sequences with a uniform and low amount of missing data per individual (~20%), SNPtotree and RAxML produced the same well-resolved and accurate phylogenetic tree. The same was observed for nested clades within clade C1b1 with low amounts of missing data and high BS values (Figure 5B,C), which were colored purple and green (according to Appendix A). For sequences with higher amounts of missing data, SNPtotree showed higher resolution compared to ML (black-colored branches in Figure 5B,C).

In both generated SNP phylogenies (Figure 5B,C), some SNPs were located in unexpected positions of the tree, e.g., S10738.2 (clade C2b1a2a1a2) and Y148127 (clade C2a1a3a6) were located within the C1b1 clade (Figure 5B,C). These may be recurrent mutations in all individuals or sequencing errors that were not filtered out since they did not cause contradictory relationships between markers or ambiguous positions.

Generally, we were able to show very similar or even equal SNP phylogenies resulting from ML-constructed trees and SNPtotree when the sequences had a small number of missing bases. However, SNPtotree requires less manual work, which reduces the risk of human error. For sequences with a moderate to high number of missing bases, SNPtotree presented a better-resolved SNP phylogeny.

## 4. Conclusions

To date, there is no straightforward software available that sorts variants of non-recombining DNA into their hierarchical order. When the phylogeny of linked SNPs is of interest, maximum likelihood calculations and manual sorting of SNPs are often performed [44,45]. However, high numbers of missing data can complicate the data analysis. SNPtotree enables reliable sorting of SNPs into a hierarchical phylogenetic tree, even in the presence of high numbers of missing data, since it is based on pairwise variant comparisons that allow to extract all available information between variants. Thus, sequencing datasets of different qualities and completeness can be combined so that SNPtotree can be used to maintain or create phylogeny databases. These databases contribute to the understanding of populations, species, and their respective histories by finding substructures within or between populations. The characterization of populations may help to assign individuals to populations, which is applicable in forensic genetics or in pathogen analysis. The latter allows an analysis of the development of pathogens in epidemics, like those of the haploid SARS-CoV-2 variant genomes, particularly when considering the geographic origin of variants. This information is also relevant for migration pattern analyses in animals, which may reveal information on an individual’s genetic genealogy and the evolution of a species. While SNPtotree requires prior knowledge of the analyzed species, this is worth adjusting in a future version, where sequencing data of the studied species and a reference sequence of any ancestral species should be accepted to extend the applicability of SNPtotree to rather unexplored species.

## Figures and Tables

**Figure 1 genes-14-01837-f001:**
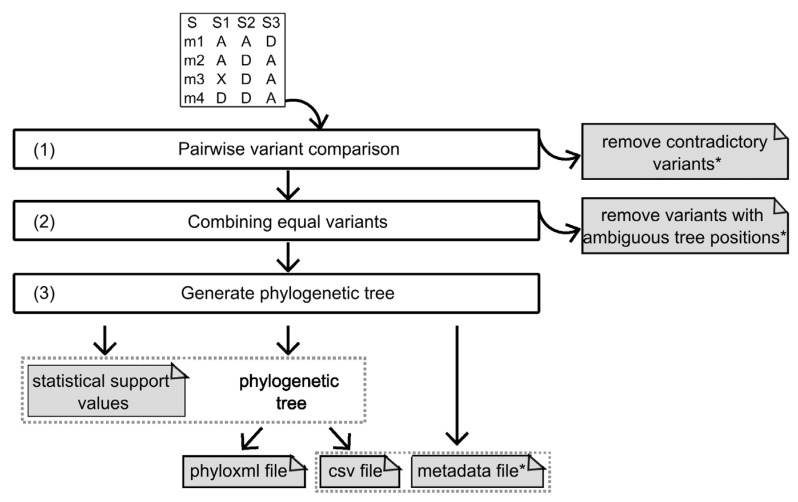
Workflow of the SNPtotree algorithm. An input file of the ancestral (A) or derived (D) allelic states of the polymorphic sites is required. Missing data should be indicated using an “X”. The header row represents the individuals’ labels (S1, S2, and S3), and the first column represents the variant names (m1, m2, m3, and m4). The algorithm consists of three steps: first (1), all pairs of variants are compared to each other to predict the pairwise relationships. Variants with contradictory relationships are removed. Second (2), variants that are not separable are predicted to be equal. During the process of finding equal variants, variants with ambiguous positions in the tree are removed. Finally (3), the hierarchical variant order is inferred, and the phylogenetic tree is generated. Additional output files (see Section 2) provide the statistical support values for each variant in the tree, and metadata specifies the sequences carrying the respective variants in the different branches of the csv output tree. Optional output files are marked with an asterisk.

**Figure 2 genes-14-01837-f002:**
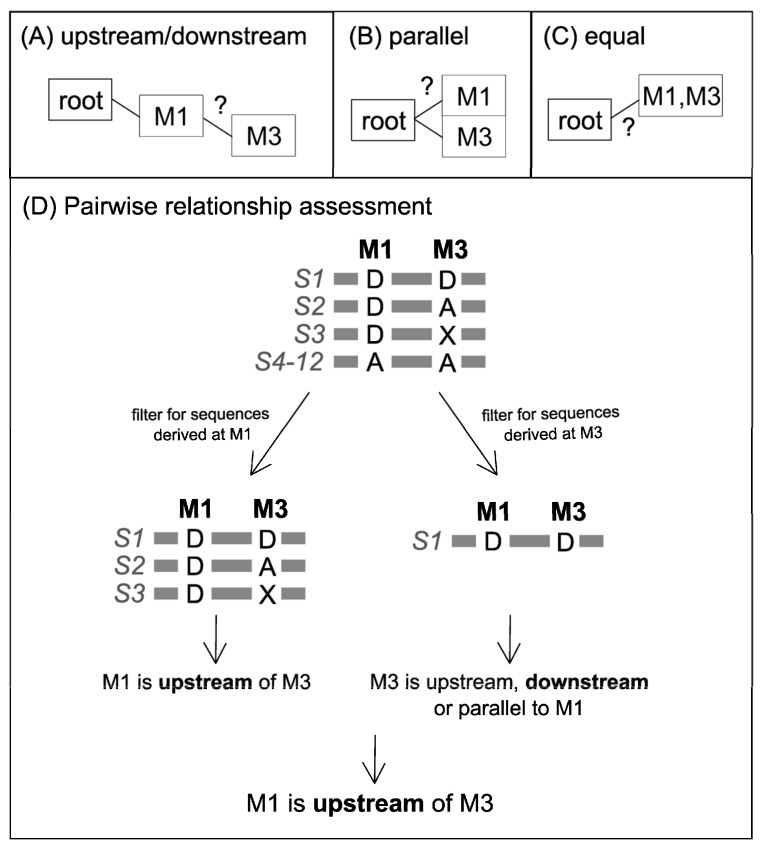
To determine whether variants M1 and M3 are (**A**) upstream/downstream, (**B**) parallel, or (**C**) equal to each other, (**D**) their pairwise relationships are assessed in a two-way comparison. Among all sequences (S1–S12), only those sequences that have a derived allelic state for M1 (or M3 in the second comparison) are considered. All observed allelic states of the remaining variant M3 (or M1) are documented, and the resulting relationships are compared. The consensus relationship defines the final pairwise relationship between variants M1 and M3.

**Figure 3 genes-14-01837-f003:**
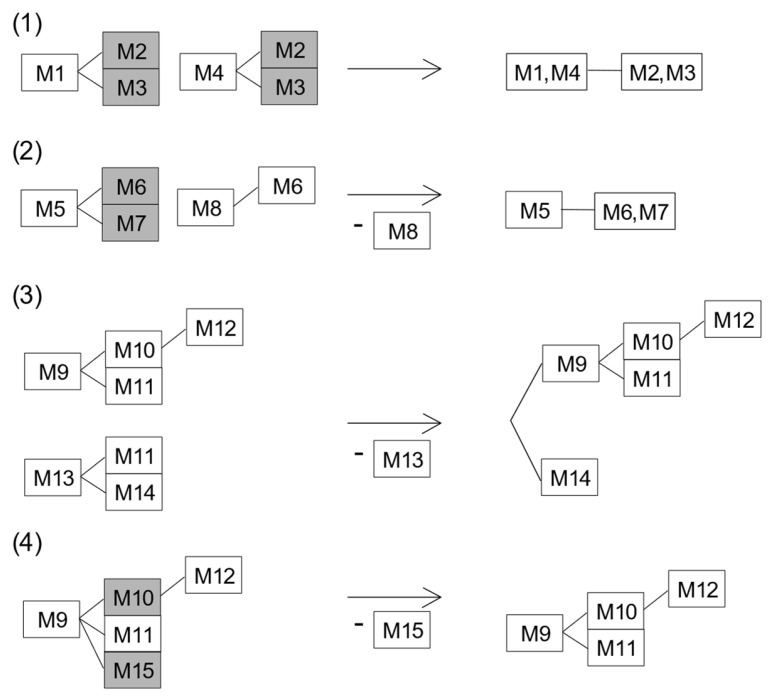
Examples of how SNPtotree combines equal variants and removes variants with ambiguous positions in the tree caused by a lack of representative data, e.g., due to missing information. The relationships between the 15 variants (M1 to M15) are based on the dataset in Table 1. Variants reported in the same sequence but presented in parallel branches are highlighted in gray boxes. In the first example (**1**), M1 and M4 are always found in the same allelic state in the sequences where both variants have been typed. Since M1 and M4 have the same downstream variants (M2 and M3), SNPtotree will consider M1 and M4 as equal. Furthermore, SNPtotree universally joins all branch tip variants (i.e., without any downstream variants) that are sharing their immediate upstream variant to groups of “equal” variants. Thus, M2 and M3 are considered equal as well. In the second example (**2**), M5 and M8 share one of their downstream variants (M6), and M5 has the unique downstream variant (M7). To avoid double entries, the upstream variant with fewer downstream variants (M8) is removed if its relationships to the residual variants (M5, M7) are unknown. In the third example (**3**), variants M9 and M13 share the downstream variant M11. Additionally, M9 and M13 have unique downstream variants (M10, M12, and M14). However, there is insufficient information connecting the two subtrees. To avoid the introduction of incorrect phylogenetic relationships, SNPtotree removes variants with several possible positions in the tree (M13). The final example (**4**) presents a rule for variants (M10 and M15), which were reported in the same state in all sequences but did not share any downstream variants. In this example, M10, M12, and M15 are downstream of M9 (Table 1). However, the relationship between M15 and the other variants is unknown because of missing data from M15 in some sequences. SNPtotree removes M15 to maintain maximum depth in the tree.

**Figure 4 genes-14-01837-f004:**
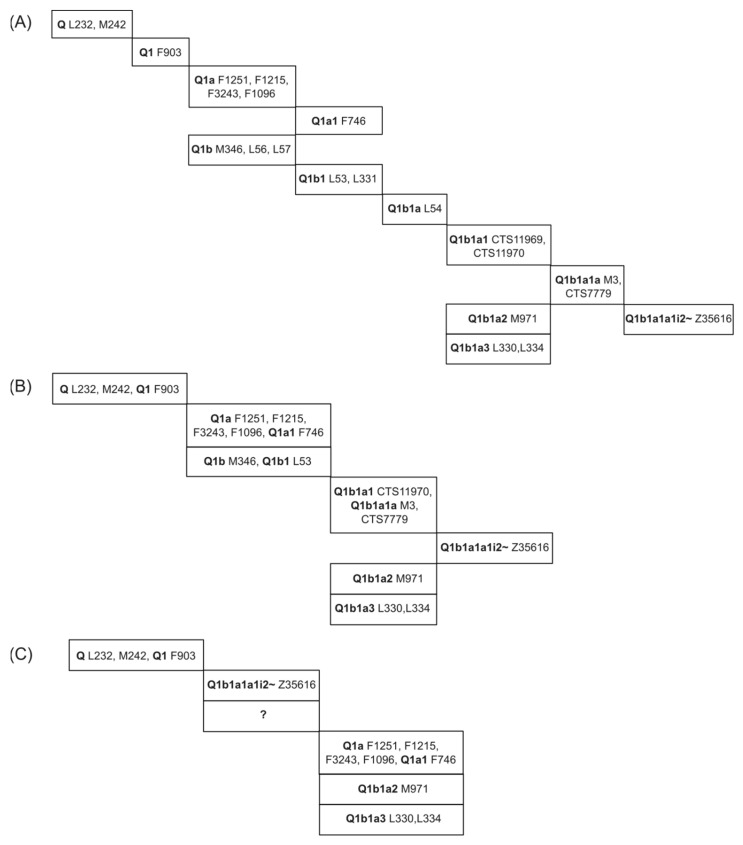
(**A**) True phylogeny of the 22 variants given in testdata 1 (taken from the ISOGG Y-DNA Haplogroup Tree 2019–2020). Clade names, which precede the SNP names, are highlighted in bold. Please note that a speciation event results in a split into at least two sister lineages. Testdata 1 only contained a small subset of clade Q lineages, and only these were presented here. (**B**) Phylogeny of clade Q SNPs and lineages resulting from the SNPtotree analysis. (**C**) Phylogeny of clade Q SNPs and lineages resulting from ML tree construction and manual sorting.

**Figure 5 genes-14-01837-f005:**
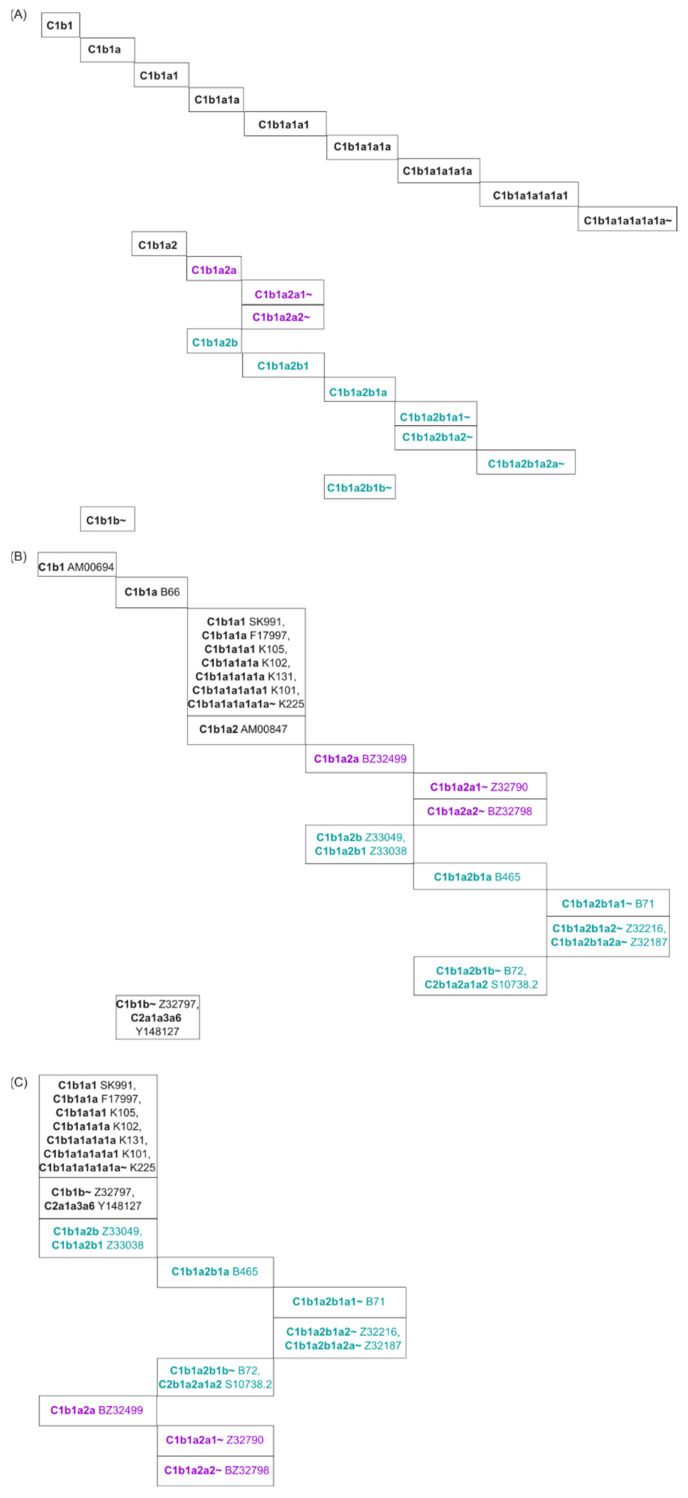
(**A**) True phylogeny of the 21 subclades determined by all known SNPs from testdata 2 within the human Y-chromosomal clade C1b1 (phylogenies taken from ISOGG Y-DNA Haplogroup Tree 2019–2020). The SNP names were omitted to present the tree in a simple way. (**B**) Phylogeny of clade C1b1 SNPs and subbranches resulting from SNPtotree analysis. (**C**) Phylogeny of clade C1b1 SNPs and subbranches resulting from ML tree construction. The nested clades with high BS values composed of sequences with low missing data were colored purple or green, corresponding to the clades in Appendix A.

**Table 1 genes-14-01837-t001:** Example of an input dataset with twelve sequences and 16 variants, sorted into three main branches (highlighted in gray).

		Sequences											
		S1	S2	S3	S4	S5	S6	S7	S8	S9	S10	S11	S12
Variants	M1	D	D	D	A	A	A	A	A	A	A	A	A
	M2	D	A	A	A	A	A	A	A	A	A	A	A
	M3	D	A	X	A	A	A	A	A	A	A	A	A
	M4	D	D	D	A	A	A	A	A	A	A	A	A
	M5	A	A	A	D	D	D	A	A	A	A	A	A
	M6	A	A	A	D	A	A	A	A	A	A	A	A
	M7	A	A	A	D	A	X	A	A	A	A	A	A
	M8	A	A	A	D	X	D	A	A	A	A	A	A
	M9	A	A	A	A	A	A	D	D	D	D	D	X
	M10	A	A	A	A	A	A	D	A	A	D	D	A
	M11	A	A	A	A	A	A	A	A	D	A	A	A
	M12	A	A	A	A	A	A	D	A	A	A	D	A
	M13	A	A	A	A	A	A	X	D	D	X	X	D
	M14	A	A	A	A	A	A	X	D	X	X	X	A
	M15	A	A	A	A	A	A	D	X	A	X	D	A
	M16	D	D	D	D	D	D	A	X	A	X	D	A

**Table 2 genes-14-01837-t002:** Assessment of pairwise variant relationships based on observed allelic states of variant 2 when considering sequences in which variant 1 is in the derived allelic state.

All Sequences Where Variant 1 is Derived		
Variant 1	Variant 2	Variant 1 Compared to Variant 2 Is…
D	D	equal, downstream, or upstream
D	D + A	upstream
D	A	parallel or upstream
D	X	parallel, upstream, downstream, or equal
D	D + X	equal, downstream, or upstream
D	A + X	parallel or upstream
D	A + D + X	upstream

**Table 3 genes-14-01837-t003:** Each variant’s downstream variant(s) result from the two-way pairwise comparisons and represent preliminary tree branches.

Variant	Downstream Variant(s)
M1	M2, M3
M4	M2, M3
M5	M6, M7
M8	M6
M9	M10, M11, M12, M15
M10	M12
M13	M11, M14

## Data Availability

The publicly available biological data analyzed during this study are included in Bergström et al. (2016) and Karmin et al. (2015), as referenced in the manuscript. The software is publicly available on https://github.com/ZehraKoksal/SNPtotree (accessed on 20 August 2023) with the GNU General Public License v3.0, using the programming language Python, and with the following requirements: argparse ≥ 1.1, pandas ≥ 1.1.5, and numpy ≥ 1.19.5.

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
