# Peer review of "SNPtotree—Resolving the Phylogeny of SNPs on Non-Recombining DNA"

_genes, 2023, doi:10.3390/genes14101837_

Round 1

Reviewer 1 Report

Review of the article "SNPtotree – Resolving the phylogeny of SNPs on non-recombining DNA".

My points to improve the article:

1. The Abstract is written in very general way. The Abstract should present the aim of the work more clearly and directly, along with a short summary of the results obtained.

2. In the Introduction information about the MEGA program should be added. The MEGA program is very often used in phylogeny.

3. Artificial neural networks (ANNs) are increasingly used in the reconstruction of phylogeny (reconstruction of evolution) and can provide good results also in the case of imprecise data. For this reason, information about the use of ANNs should be added in the Introduction.

4. The description of the proposed method presented in the article is not easy to understand. In my opinion, the Authors should add a block diagram of the algorithm (that is presented in the 2.1.4. Algorithm section) of the proposed method in the article.

5. In the Materials and Methods section is "SNPtotree organizes biallelic variants hierarchically by performing pairwise comparisons".

The sequences need to be aligned before comparison, but I cannot find in the article information about alignment method.

6. The new method proposed by the Authors has been evaluated in comparison with the ML method. The consensus tree generated by the ML method depends on the number of repetitions in the Bootstrap (BS) method, as well as on the set substitution model (the article only provides information that the Gamma model of rate heterogeneity was used).

Low Bootstrap (BS) values obtained using the ML method may simply result from setting too few repetitions in the Bootstrap (BS) method (or perhaps also from a suboptimal substitution model).

Surprisingly, the configuration of the consensus tree generated using the ML method also depends on the number of repetitions in the Bootstrap (BS) method.

For these reasons, the Authors should present and discuss the results for a larger number of repetitions in the Bootstrap (BS) method, as well as for different substitution models.

7. The References section should be corrected, i.e. all items have to be written in the MDPI style.

Reviewer 2 Report

Dear Editors,

The reviewed study entitled "SNPtotree – Resolving the phylogeny of SNPs on non-recom-2 bining DNA" aimed to compare the performance of SNPtotree, a new software tool, with the traditional maximum likelihood (ML) tree construction method in resolving the phylogenetic hierarchy of single nucleotide polymorphisms (SNPs) on non-recombining DNA. They used two datasets, testdata 1 and testdata 2, to evaluate the accuracy and effectiveness of SNPtotree.

Overall, the study appears to be well-structured, and the methodology is promising for addressing the research problem. Therefore, the study deserves to be published in Genes periodical. Detailed review can be found in the attached file.

Best regards

Dear Editors,

The reviewed study entitled "SNPtotree – Resolving the phylogeny of SNPs on non-recom-2 bining DNA" aimed to compare the performance of SNPtotree, a new software tool, with the traditional maximum likelihood (ML) tree construction method in resolving the phylogenetic hierarchy of single nucleotide polymorphisms (SNPs) on non-recombining DNA. They used two datasets, testdata 1 and testdata 2, to evaluate the accuracy and effectiveness of SNPtotree.

Overall, the study appears to be well-structured, and the methodology is promising for addressing the research problem. Therefore, the study deserves to be published in Genes periodical. Detailed review can be found in the attached file.

Best regards

Reviewer 3 Report

The authors present a software for phylogenetic analysis of SNP data. However, their method has substantial drawbacks, and it is not described clearly in the manuscript. I doubt that it will be of use in any real world cases. Briefly, in most cases there will be more than two possible alleles, and it will not be possible to define which of them is "ancestral" and which is "derived". Furtheron, the algorithm does not determine branch lengths. I just understood towards the end of the manuscript that the algorithm does not even determine a tree of the *samples* but just orders the *variant sites*, which is probably of no interest to researchers. I will address my criticism in the order of the paper:

l. 36: The root is the root, not a "global MRCA".

l. 38: "time of accumulation of these variants can be determined": this is not true; e.g. Fig. S4 clearly shows that the branches are equally spaced. The *order* of mutations, not their time is determined by their algorithm.

l. 47f: "fewest number of character changes, irrespective of the nucleotide substitution" - nucleotide substitutions *are* character changes.

l. 109ff: These command line parameters are a complete mess. why is there two times "path_output_folder" and two times "path_output_data_folder"? There are spaces in the parameters.

Fig. 1: Who determines "A" and "D" for each variant? What happens if there is a third allele? 2.1.4.1, what is "horizontal relationship"? That would mean that arrays of SNPS for all samples are compared between variant sites? That sounds strange. So, you compute relationships between variant positions, not between samples. Why is this step done first? What are "contradictory variants"? What are "variants with ambiguous tree positions"? How can you remove them before even constructing the tree? Why is the tree not output in any common format like Nexus or Newick? What are variants with "contradictors relative relationships" (l. 120f)? What does it mean that two variants are "not separatable" (l. 121)? How exactly is the hierarchical variant order inferred (l. 123)?

l. 152f: "variants of each tree row are returned in the respective row of the first coloum" - not comprehensible.

l. 166-170: I do not understand the whole paragraph. Do I get it right that you compare between *variant posoitions*, not between samples here? If you compare betwen variants, then of course "A" is upstream and "D" is downstream. "in sequences with a derived allelic state of variant 1 [...] variant 1 cannot be ancestral" - that sounds obvious.

l. 171ff: "...located in separate branches...", "located in the same branch" - you did not calculate the tree yet, how do you know about branches?

l. 178-182: what exactly are you comparing here and how?

Tab. 1: How can one variant always be "D"? That would be, in all sequences? How about "D+A", does that mean both nucleotides appear in any sequewnce? You do not care about how often? l. 185-190 what is "first comparison", "second comparison" between M1 and M2?

l. 203: describe exactly what a "contradictory finding" is. In this case, are *both* variants removed? How often does that occur, how many variants are removed in a typical use case?

l. 205 "initial tree is constructed" how?

l. 209 what is a "vertical comparison"?

l. 244-245 "infer the most parsimonious hierarchical order" how?

l. 251-258 why does the data set with 51 individuals have only 22, but the data set with 46 individiuals 4348 variant positions? How many triallelic variants did occur in the data sets, what did you do with them? How did you determine, for each variant position, which allele is "A" and which is "D"?

l. 260 "SNPs with reported phylogenetic relationships" what does that mean?

l. 267 "C1a splits into C1a1, C1a2, and C1a3" how can you have three nodes from biallelic variants? The naming convention is not clear. How can you name variants before constructing the tree?

l. 281f: where does the publicly available phylogeny come from? Why is it the gold standard?

l. 286: where are the seven "layers" in the figure? 5 of 22 variants have been romoved??

l. 290: what are certainty values?

Fig. 1: Why do you not present a "normal" phylogeetic tree? How can, e.g., Q1b1a1a have two ancestors?

Fig. S5: How did you decide where to put the extra block of B100 and B99?

l. 321 what are layers?

What is the run time of the algorithm?

Round 2

Reviewer 1 Report

The Authors have correctly addressed all my concerns and comments. Now, the article is much better and in my opinion it can be published in Genes.

Reviewer 3 Report

The manuscript improved a lot and is now fine for publication.